# Impact of voluntary termination of pregnancy on female sexual function: A french monocentric longitudinal study

Léa Rouchou[1]*, Dounia Baïta[1], Sophie Regueme[2], Claude Hocke[1]

1 Department of Gynecological Surgery, Medical Gynecology, and Reproductive Medicine, Aliénor d'Aquitaine Center, Pellegrin University Hospital, Bordeaux, France, 2 Bordeaux University Hospital, Bordeaux, France

* lea.rouchou@chu-bordeaux.fr

## Abstract

### Background

Voluntary termination of pregnancy (VTOP) is common in France, yet its impact on female sexual function remains incompletely understood. Previous studies often reported changes in Female Sexual Function Index (FSFI) scores without validated diagnostic thresholds, limiting clinical interpretation. This study aimed to assess the prevalence and evolution of sexual dysfunction over six months following VTOP, and to explore factors associated with persistent dysfunction using a validated FSFI threshold.

### Methods

We conducted a prospective, longitudinal study enrolling 186 adult women undergoing VTOP at Bordeaux University Hospital (Feb–May 2023). Sexual dysfunction was assessed at inclusion and at 1, 3, and 6 months post-VTOP using the validated FSFI (dysfunction defined as score ≤26.55). Additional medical and socio-demographic data were collected via self-administered questionnaires. Logistic mixed-effects models evaluated time effects and factors associated with dysfunction. The primary outcome was the prevalence of sexual dysfunction over time.

### Results

Among participants completing follow-up (n = 47 at 6 months), sexual dysfunction prevalence decreased from 61.3% at baseline to 36.2% at 6 months (aOR = 0.17; 95% CI, 0.09–0.29). Median FSFI scores increased from 24.7 to 28.5 (p = 0.012), particularly in desire, arousal, and orgasm domains. Being single (aOR = 2.17; 95% CI, 1.09–4.34) and reporting psychological symptoms were independently associated with persistent dysfunction.

**Data availability statement:** All relevant data are within the paper and its Supporting Information files.

**Funding:** The author(s) received no specific funding for this work.

**Competing interests:** The authors have declared that no competing interests exist.

## Discussion

Among women completing follow-up, sexual function appeared to improve over six months following pregnancy resolution. Integrating psychological and relational support into post-VTOP care may help optimize recovery. The high attrition and monocentric design limit generalizability and warrant cautious interpretation.

## Introduction

Voluntary termination of pregnancy (VTOP) is a central component of reproductive health in France, with 243,623 procedures performed in 2023, [1] a number that has remained stable since legalization. Physical risks are low in regulated settings, and psychological effects, primarily studied in the early 2000s, are generally transient. [2–4] Sexual health, defined by the World Health Organization (WHO) as a state of physical, mental, and social well-being in relation to sexuality, is closely linked to quality of life. [5] Disorders affecting desire, arousal, orgasm, or causing pain (DSM-5) are common yet under-researched in women. [6] In France, 30–50% of women report experiencing some form of sexual dysfunction, [7] highlighting its prevalence and significant impact.

Despite its clinical importance, sexual function following VTOP remains insufficiently studied. Early investigations, mostly conducted in the 2000s and relying on self-reported or descriptive data, suggested that VTOP could temporarily affect sexual function. [2,4,8–12] However, these studies were limited by non-standardized assessment tools and short follow-up periods, reducing their reliability. More recent research using the validated Female Sexual Function Index (FSFI) [13] reported conflicting results: some studies noted improvement in scores six months post-VTOP without evaluating dysfunction prevalence, [14] while others noted procedure-specific differences. [15] Critically, these studies rarely applied the FSFI diagnostic threshold, limiting clinical relevance.

In France, psychological support is routinely offered as part of VTOP care, but sexual health is rarely addressed. We hypothesized that sexual dysfunction would decrease over six months following VTOP among women completing follow-up, and that socio-demographic and psychological factors would be associated with persistent dysfunction. To test these hypotheses, this study assessed the prevalence and evolution of sexual dysfunction within six months following VTOP using a validated FSFI threshold, and explored factors associated with persistent dysfunction through longitudinal analyses.

## Methods

### Study design and participants

This prospective, longitudinal, descriptive, monocentric study was conducted at the Family Planning Unit of Bordeaux University Hospital, France. Eligible participants were adult women (≥18 years) undergoing medical or surgical VTOP and covered by health insurance. Exclusion criteria included non-French-speaking, illiteracy, women

subjected to legal protection measures or with psychiatric comorbidities, unable to complete follow-up or women whose VTOP was indicated as a result of sexual assault. Enrollment was conducted from February 27, 2023, to May 24, 2023.

## Follow-up and data collection

Participants were included on the day of hospitalization, either for misoprostol administration in an outpatient setting (medical VTOP) or for surgical procedures (surgical VTOP).

Participants completed the FSFI and a study-specific questionnaire at inclusion and at 1.5, 3.5, and 6.5 months post-VTOP. The timing accounted for the recommended 15-day post-procedure abstinence period.

The study concluded once the final questionnaires were completed at 6 months post-VTOP. To minimize loss to follow-up, reminder SMS messages were sent to participants who had not completed the final questionnaires between February 5 and February 10, 2024.

The FSFI, a validated 19-item tool in French [16], designed to assess female sexual function across six domains: desire, arousal, lubrication, orgasm, satisfaction, and pain, based on experiences over the preceding four weeks. It is widely used in clinical and research settings due to its robust psychometric properties and ability to capture multifaceted aspects of sexual health. A score ≤26.55 indicates dysfunction, validated through psychometric analysis [13].

Using the FSFI score as a binary outcome (sexual dysfunction vs. no dysfunction) is consistent with clinical practice guidelines, as it allows for a clear, standardized distinction between individuals experiencing sexual dysfunction and those who are not.

The second questionnaire was a study-specific self-report tool, comprising 11 items. It addressed sexual activity during the year prior to the VTOP (including frequency of sexual activity and number of partners), marital status, history of violence (including sexual, psychological, verbal, and physical abuse), as well as self-reported psychological symptoms such as fatigue, anxiety, sadness, and guilt.

During follow-up, participants completed both the FSFI and a custom follow-up questionnaire at each designated time point. The follow-up questionnaire (comprising 6–7 items) assessed sexual activity (including frequency, number of partners, and reasons for changes in sexual activity), contraception use, self-reported psychological symptoms (e.g., fatigue, anxiety, sadness, guilt), and discussions about sexuality during follow-up consultations.

All data were collected via a secure online platform and pretested for clarity and acceptability.

Demographic and clinical data were retrieved from hospital records.

All data were anonymized, coded, and entered electronically by the IT department of Bordeaux University Hospital.

## Sample size

Given the limited literature on this topic, the sample size was determined empirically, taking into account the findings of two closely related studies (Morotti et al. [15], 2015; Limoncin et al. [14], 2014) and the service's recruitment capacity. A two-month inclusion period was planned, during which approximately 200 participants were expected to be enrolled based on service activity data. While this sample size estimation is appropriate for an exploratory study, it is important to acknowledge that this approach may limit confidence in non-significant findings, particularly for secondary outcomes and adjusted analyses conducted at later time points. As such, the results should be interpreted cautiously, and null findings in exploratory analyses should not be overinterpreted.

## Statistical methods

Categorical variables were summarized as frequencies and percentages, and continuous variables as medians with interquartile ranges (IQR).

The primary outcome was the presence of sexual dysfunction, defined as an FSFI total score below the clinical threshold. Logistic mixed-effects models were used to assess sexual dysfunction over time, with the effect of time modeled quadratically to account for non-linear trends. Logistic mixed-effects models were conducted, adjusting for six factors identified from the literature: age, parity, relationship status, history of violence, and psychological symptoms assessed both before and after the unintended pregnancy.

Continuous FSFI scores were analyzed using linear mixed-effects models. To identify factors associated with sexual dysfunction longitudinally, mixed-effects logistic regression models with a random intercept for each participant and a random slope for time were applied, accounting for individual variability and repeated measures at baseline, 1, 3, and 6 months post-VTOP.

In addition, descriptive analyses were conducted to examine factors potentially associated with sexual dysfunction at inclusion and during follow-up. Variables identified from the literature were compared between groups using Chi-square or Fisher's exact tests, as appropriate.

Missing data were not imputed, as incomplete questionnaires were excluded and loss to follow-up was considered likely Missing Not At Random (MNAR).

As a conservative sensitivity analysis, we assumed that participants with missing follow-up FSFI data did not change their sexual function status. Women with sexual dysfunction at the last observed time point were assumed to remain dysfunctional thereafter, while those without dysfunction were assumed to remain unaffected. A mixed logistic regression model was then re-fitted.

Data were analyzed using Stata software (version 16.1), with statistical significance set at p < 0.05.

### Ethical considerations

VTOP-SImpact study (NCT05688228) was conducted closely with international guidance including the Good Clinical Practices and the Declaration of Helsinki. Accordingly to French national laws the study was approved by a national Ethics Committee (CPP Sud-Ouest et Outre-Mer III) and the competent authority (ANSM) has been informed. The processing and use of data collected as part of this study have been done in accordance with the General Data Protection Regulation (GDPR – EU 2016/679) and complies with Reference Methodology (MR-003) of French Data Protection laws in force. Informed consent was obtained orally from all participants, in accordance with ethics committee guidelines.

## Results

### Study population

Among the 344 women admitted to the Family Planning Unit during the inclusion period, 186 participants were included (Fig 1).

At the 1-month follow-up, 80 participants (43.0% of the initial sample) completed the questionnaires. At 3 and 6 months, the number of participants decreased to 48 (25.8%) and 47 (25.3%), respectively. Additionally, 24 participants (12.9% of the initial sample) completed all assessments.

Table 1 presents the baseline sociodemographic and clinical characteristics of the study population. Participants had a median age of 25 years (IQR: 22–32). Most were in a relationship (81.7%), childless (65.1%), and underwent surgical VTOP (66.1%). One-third (28.0%) reported a history of VTOP. Over half (57.0%) reported a history of violence, with sexual violence accounting for nearly one-third of participants (29.0%). Self-reported psychological symptoms were frequent both before (85.0%) and after (91.4%) pregnancy discovery. Sexual symptoms were reported by 72.6%, with reduced desire affecting 55.9%.

Participant characteristics remained generally consistent across follow-up; however, study completers were more likely to report sexual violence (p = 0.026) and pre-pregnancy anxiety (p = 0.002) (S1 Table). No significant differences

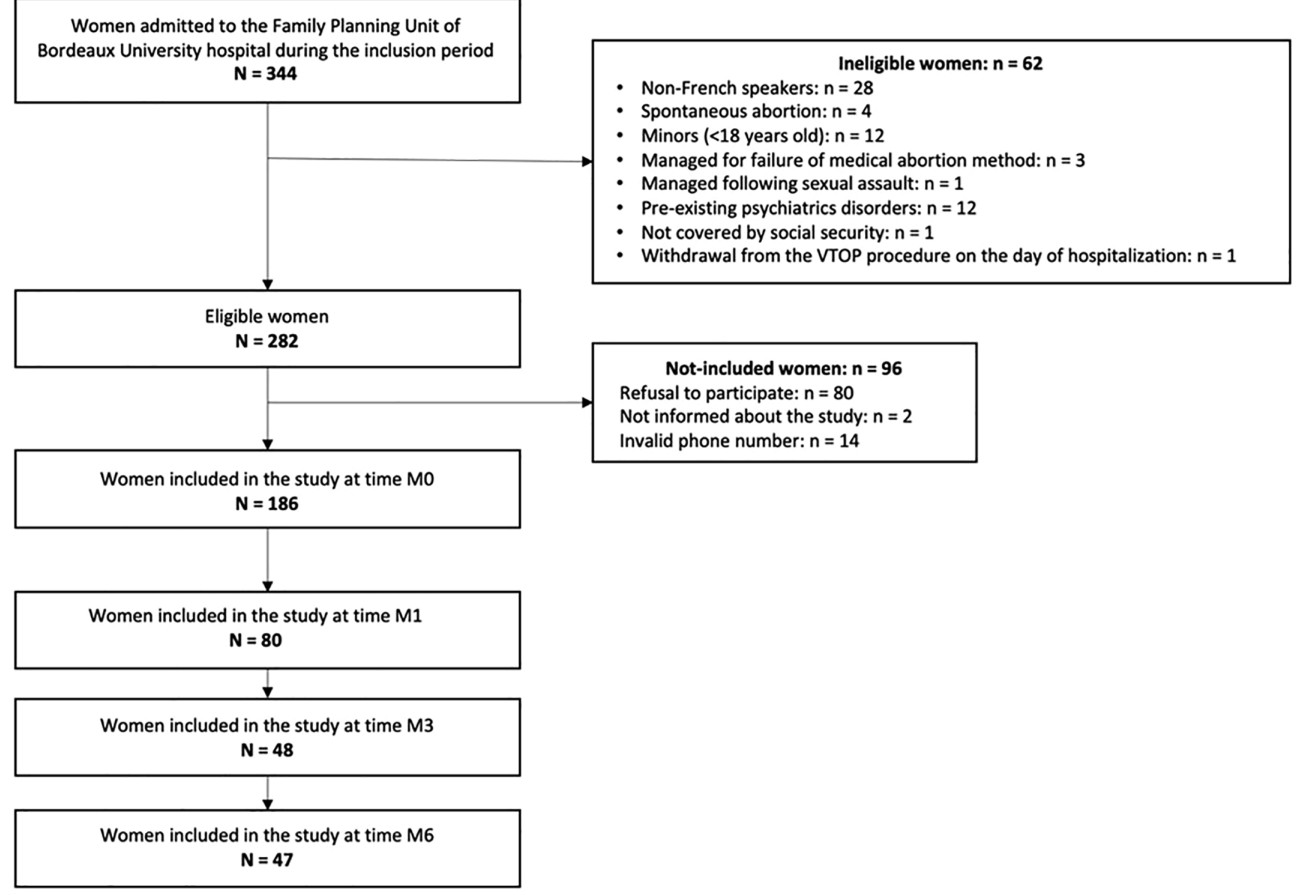

**Fig 1. Selection of the study population.**

were found between respondents and non-respondents at 6 months, with the exception of a slightly higher prevalence of reported physical violence among respondents (p = 0.020) (S2 Table).

## Sexual activity during follow-up

At baseline, most participants reported being sexually active before the discovery or the pregnancy (S3 Table). Sexual activity persisted throughout follow-up, although a slight decrease in frequency was observed over time. Fear of a new pregnancy was the most frequently reported reason for reduced sexual activity during follow-up.

## Prevalence of sexual dysfunction at baseline and post-VTOP

Sexual dysfunction prevalence among participants completing each follow-up was as follows (Fig 2): 48.8% (39/80; 95% CI, 37.4–60.2) at 1 month, 45.8% (22/48; 95% CI, 31.2–60.2) at 3 months, and 36.2% (17/47; 95% CI, 22.7–51.5) at 6 months post-VTOP. Logistic mixed-effects modeling showed a significant reduction in sexual dysfunction over time (OR 0.176; 95% CI, 0.100–0.311), which remained significant after adjusting for age, parity, relationship status, history of violence, and self-reported psychological symptoms (adjusted OR 0.167; 95% CI, 0.094–0.292; p < 0.0001).

In a conservative sensitivity analysis assuming no improvement in sexual dysfunction among participants with missing follow-up data, Logistic mixed-effects after adjusting for age, parity, relationship status, history of violence, and

**Table 1. Baseline characteristics of the participants (n = 186).**

| Baseline characteristics | Frequency (%) |
|---|---|
| **Age** *Median (IQR)* | *25 (22–32)* |
| *18–25 years* | 99 (53.2) |
| *26–35 years* | 63 (33.9) |
| *>35 years* | 24 (12.9) |
| **Parity** *Median (IQR)* | *0 (0–1)* |
| *No children* | 121 (65.1) |
| *At least one child* | 65 (34.9) |
| **Number of previous VTOPs** | |
| *None* | 134 (72.0) |
| *At least one* | 52 (28.0) |
| **VTOP method performed** | |
| *Medical* | 63 (33.9) |
| *Surgical* | 123 (66.1) |
| **Relationship status prior to the procedure** | |
| *Single* | 34 (18.3) |
| *In a relationship* | 152 (81.7) |
| **History of violence at least once in a lifetime** | 106 (57.0) |
| *History of sexual violence* | 54 (29.0) |
| *History of physical violence* | 63 (33.9) |
| *History of psychological violence* | 94 (50.5) |
| **Self-reported psychological symptoms before discovering the pregnancy** | 158 (85.0) |
| *Fatigue* | 152 (81.7) |
| *Sadness* | 60 (32.3) |
| *Anxiety* | 68 (36.6) |
| *Guilt* | 36 (19.3) |
| **Self-reported psychological symptoms following the discovery of the pregnancy** | 170 (91.4) |
| *Fatigue* | 161 (86.6) |
| *Sadness* | 97 (52.1) |
| *Anxiety* | 88 (47.3) |
| *Guilt* | 65 (35.0) |
| **Sexual symptoms after discovering the pregnancy** | 135 (72.6) |
| *Desire disorders* | 104 (55.9) |
| *Arousal disorders* | 71 (38.2) |
| *Lubrification issues* | 30 (16.1) |
| *Orgasm disorders* | 30 (16.1) |
| *Satisfaction issues* | 34 (18.3) |
| *Dyspareunia* | 21 (11.3) |

Data are presented as n (%) unless otherwise specified as median (IQR). Percentages are calculated based on the total study population at inclusion (n = 186).

VTOP: Voluntary Termination of Pregnancy.

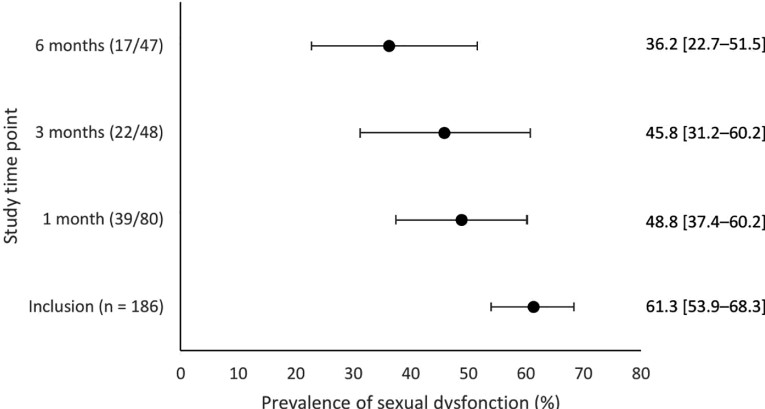

**Fig 2. Prevalence of sexual dysfunction at each study time point.** Sexual dysfunction prevalence is presented as a percentage for each study time point, with 95% confidence intervals represented by horizontal error bars.

self-reported psychological symptoms modeling showed a significant reduction in sexual dysfunction over time (adjusted OR 0.84; 95% CI, 0.75–0.95, p = 0.007).

### Sensitivity analysis

Sensitivity analysis (Table 2) using a linear mixed model revealed significant improvement in FSFI scores over time (p = 0.012), with the median increasing from 24.7 (IQR: 18.5–29.0) at baseline to 28.5 (IQR: 24.2–30.5) at 6 months (Table 2). Statistically significant improvements were observed in desire (p < 0.001), arousal (p = 0.020), and orgasm (p = 0.018) subdomains. These improvements exceeded the minimal clinically important difference threshold of 0.5–1 point. Although statistically significant, these improvements should be interpreted cautiously, as they may reflect both natural recovery over time after pregnancy resolution and potential effects of the VTOP procedure. The high attrition rate further limits the interpretation.

These analyses are exploratory and the results should be treated as preliminary due to the limited sample size and attrition over follow-up.

### Factors associated with sexual dysfunction: Multivariable mixed logistic regression analysis

Table 3 presents the univariate and multivariate mixed-effects model analyses of factors associated with sexual dysfunction. Single women were significantly more likely to report sexual dysfunction (aOR 2.17; 95% CI, 1.09–4.34; p = 0.027). No significant associations were found for other factors, including history of violence, self-reported psychological symptoms, age, parity, VTOP method (medical vs. surgical), or contraceptive use (all p > 0.05).

Given the exploratory nature of these analyses and the limited statistical power due to the attrition rate, non-significant results should be interpreted with caution.

### Factors at inclusion and follow-up associated with sexual dysfunction

Secondary analyses identified additional trends (S4 Table and S5 Table). At inclusion, women with sexual dysfunction were more likely to report psychological symptoms prior to discovering their pregnancy, including sadness (p = 0.008) and guilt (p = 0.024), with no significant associations for age, parity, or history of violence.

At 1 month, single women and those with dysfunction at inclusion were more likely to report sexual dysfunction (p = 0.042). At 3 months, sadness (p = 0.004) and anxiety (p = 0.004) were more prevalent among women with dysfunction,

**Table 2. Changes in FSFI scores (total and by sexual function domain) across the study time points (baseline, 1 month, 3 months, 6 months).**

| | *Min–max* | Inclusion | 1 month | 3 months | 6 months | p |
|---|---|---|---|---|---|---|
| | | n = 180 | n = 80 | n = 48 | n = 47 | |
| **Total score** | 2–36 | 24.7 (18.5–29.0) | 26.7 (22.5–30) | 27.7 (21.1–30.3) | 28.5 (24.2–30.5) | **0.012** |
| *Desire* | 1.2–6 | 3.6 (2.4–4.2) | 4.2 (3.6–4.8) | 4.8 (3.0–5.4) | 4.8 (3.6–5.4) | **<0.001** |
| *Arousal* | 0–6 | 4.8 (2.7–5.4) | 4.8 (3.9–5.7) | 5.4 (3.8–5.7) | 5.1 (4.2–5.7) | **0.020** |
| *Lubrification* | 0–6 | 4.8 (3.3–6) | 5.4 (4.2–6) | 5.4 (3.6–6) | 5.4 (4.2–6) | 0.099 |
| *Orgasm* | 0–6 | 4 (2–5.6) | 4.4 (3.2–5.6) | 4.8 (2.6–5.6) | 4.8 (3.2–6) | **0.018** |
| *Satisfaction* | 0.8–6 | 4.8 (3.2–6) | 5.2 (3.6–6) | 5.2 (3.4–6) | 5.6 (3.2–6) | 0.215 |
| *Pain* | 0–6 | 2.8 (2.8–3.2) | 2.8 (2.8–3.2) | 2.8 (2.8–3.2) | 2.8 (2.8–3.2) | 0.957 |

Data are presented as median (IQR).

P-values were derived using linear mixed models with a random slope for time, adjusted for age, parity, relationship status, history of violence, and self-reported psychological symptoms both before and after the unintended pregnancy.

The "Min-Max" category represents the theoretical minimum and maximum values that could be obtained for each domain.

**Table 3. Univariate and adjusted analysis factors associated with sexual dysfunction using mixed-effects models (secondary objective).**

| Variables | Univariate analysis | | Adjusted analysis | |
|---|---|---|---|---|
| | OR [95% CI] | *p* | aOR [95% CI] | *p* |
| **Age** | 0.981 [0.942–1.021] | 0.340 | 0.996 [0.948–1.047] | 0.890 |
| **Parity** | | 0.212 | | 0.577 |
| *No children* | Ref. | | Ref. | |
| *At least one child* | 0.846 [0.651–1.100] | | 0.906 [0.639–1.283] | |
| **Number of previous VTOPs** | | 0.463 | | 0.788 |
| *None* | Ref. | | Ref. | |
| *At least one* | 0.795 [0.431–1.467] | | 0.921 [0.503–1.683] | |
| **VTOP method performed** | | 0.274 | | 0.361 |
| *Medical* | 1.347 [0.790–2.296] | | 1.299 [0.741–2.276] | |
| *Surgical* | Ref. | | Ref. | |
| **Relationship status prior to the procedure** | | 0.024 | | **0.027** |
| *Single* | 2.278 [1.117–4.649] | | 2.174 [1.090–4.337] | |
| *In a relationship* | Ref. | | Ref. | |
| **History of violence at least once in a lifetime** | 1.179 [0.721–1.928] | 0.512 | 1.138 [0.678–1.907] | 0.625 |
| **Self-reported psychological symptoms before discovering the pregnancy** | 1.483 [0.711–3.095] | 0.293 | 1.486 [0.701–3.149] | 0.301 |
| **Self-reported psychological symptoms following the discovery of the pregnancy** | 1.302 [0.494–3.430] | 0.593 | 1.191 [0.451–3.144] | 0.724 |

Data are presented as OR (odds ratio) and 95% CI (confidence interval).

aOR values are adjusted for age, parity, relationship status, history of violence, and self-reported psychological symptoms both before and after the unintended pregnancy.

P-values were calculated using linear mixed models with a random slope for time, both unadjusted and adjusted for age, parity, relationship status, history of violence, and self-reported psychological symptoms before and after the unintended pregnancy.

and dysfunction at previous time points predicted dysfunction. At 6 months, guilt (p = 0.035) and fatigue (p = 0.047) showed trends, with dysfunction at 3 months (p < 0.0001) and absence of prior VTOP (p = 0.0015) associated with dysfunction.

These findings are exploratory in nature and should be interpreted cautiously. Given the limited sample size and follow-up attrition, non-significant associations should not be overinterpreted and further research with larger sample sizes is needed.

## Discussion

Among women completing follow-up, sexual dysfunction appeared largely transient over six months, with significant improvement within this period. Using a logistic mixed-effects model, we observed an inverse association between sexual dysfunction and time following pregnancy resolution, indicating progressive recovery of sexual function among women completing follow-up. Changes in FSFI-defined dysfunction prevalence may not fully capture the nuances of sexual function recovery. Specifically, the distinction between FSFI-defined dysfunction prevalence and changes in continuous FSFI scores provide a more detailed understanding of the improvements over time. Continuous FSFI score changes also indicate significant improvement, particularly in the desire, arousal, and orgasm subdomains. However, it is important to note that while this improvement could be partly related to the VTOP procedure, it may also reflect the resolution of the unintended pregnancy and natural temporal regression. The low baseline FSFI scores may partly reflect a transient reduction in sexual activity around the period of unintended pregnancy and VTOP, as well as the associated emotional and psychological stress. However, this association cannot be formally established due to the absence of a true pre-pregnancy baseline. These findings are consistent with recent studies suggesting sustained improvement in sexual function after pregnancy resolution [14,15].

At baseline, 61.3% of participants exhibited sexual dysfunction, higher than the 49% reported in a 2017 Italian study using the FSFI, [14] but similar to the 63.7% reported by Dundar et al. [11] By six months post-VTOP, the prevalence decreased to 36.2%, aligning with prior reports of 31–34% [2,14]. The higher baseline rates compared to the general population in France (13–38%) [17] and in Europe (6–13%) [18] likely reflect the acute psychological stress of unintended pregnancy.

The mechanisms underlying post-VTOP sexual dysfunction appear multifactorial. While temporary physical and hormonal effects may contribute, self-reported psychological symptoms —including guilt, sadness, anxiety, and fatigue—play a more prominent role. Relationship stability was protective, highlighting the importance of social and emotional support in sexual recovery. These findings align with previous research, suggesting that post-VTOP sexual dysfunction is not solely a direct consequence of the procedure but rather reflects broader psychological and emotional health challenges [3,19].

### Strengths and limitations

Strengths of this study include its prospective longitudinal design, use of the validated FSFI threshold for sexual dysfunction, and the inclusion of contemporary French VTOP patients. Limitations include its monocentric design, potential selection bias with a higher proportion of surgical procedures, and attrition leading to possible overestimation of dysfunction prevalence. Sample size was estimated empirically, which may limit statistical power for secondary outcomes and subgroup analyses. Results should be interpreted cautiously given the small number of participants completing 6-month follow-up, and the lack of a pre-pregnancy baseline measure complicates the assessment of causality. Additionally, psychological symptoms were self-reported by participants, with no use of validated scales for their assessment, which may limit the accuracy and generalizability of these findings.

### Clinical implications and future research

Findings underscore the importance of integrating sexual health into post-VTOP care. While systematic screening immediately post-VTOP may not be feasible, providing clear information on potential transient emotional and sexual changes can help women understand and manage their experiences. Guidance for those experiencing persistent distress may improve long-term outcomes.

However, caution should be taken when generalizing these findings, particularly in relation to medical VTOP performed in outpatient settings, which remains the most common approach. The findings should not be extrapolated to populations outside of the hospital setting. Future research should include multicenter studies to capture a broader range of care

settings, as well as qualitative investigations into patient experiences and perceived barriers. Ideally, baseline measures of sexual health prior to unintended pregnancy would allow more precise assessment of causality, though this remains challenging in practice.

## Conclusion

Sexual dysfunction is common among women seeking VTOP but appeared largely temporary among women completing follow-up. Recovery, when observed, appears to be influenced by psychological well-being and relationship stability. Healthcare providers should inform women about typical emotional and sexual changes and provide guidance for persistent distress, supporting optimal post-VTOP outcomes.

## Supporting information

**S1 Table. Comparative table of the baseline characteristics of respondent populations at various time point in the study.** Data are presented as n (%) unless otherwise indicated as median (IQR). Percentages are calculated based on the total study population corresponding to each study time point: inclusion (n = 186), 1 month (n = 80), 3 months (n = 48), and 6 months (n = 47). VTOP: Voluntary Termination of Pregnancy. P-values were calculated using Chi-square tests for independence, with a significance threshold set at 0.05.
(PDF)

**S2 Table. Comparative table of the characteristics of respondent populations at six months and no respondents at six months.** Data are presented as n (%) unless otherwise indicated as median (IQR). VTOP: Voluntary Termination of Pregnancy. P-values were calculated using Chi-square tests for independence, with a significance threshold set at 0.05.
(PDF)

**S3 Table. Sexual activity and reasons for reduced sexual activity during follow-up after voluntary termination of pregnancy (VTOP).** Baseline variables refer to sexual activity before the discovery of the unintended pregnancy. *Percentages are calculated only among participants reporting a reduction in sexual activity or intercourse during the previous month. Data are self-reported.
(PDF)

**S4 Table. Factors associated with sexual dysfunction at inclusion: Results from chi-square analysis.** Data are presented as n (%). Percentages are calculated based on the total study population at inclusion (n = 186). Sexual dysfunction is defined as an FSFI score ≤ 26.55. VTOP: Voluntary Termination of Pregnancy. P-values were calculated using Chi-square tests for independence, with a significance threshold set at 0.05.
(PDF)

**S5 Table. Factors associated with sexual dysfunction at 1, 3 and 6 months post-VTOP: Results from chi-square analysis.** Data are presented as n (%). Sexual dysfunction is defined as an FSFI score ≤ 26.55. VTOP: Voluntary Termination of Pregnancy. P-values were calculated using Chi-square tests for independence, with a significance threshold set at 0.05. Denominators correspond to the total population at each study time point, except for the variable "presence of sexual dysfunction," where the number of respondents varies: For the presence of sexual dysfunction at inclusion: n = 80 at 1 month; n = 48 at 3 months; n = 47 at 6 months. For the presence of sexual dysfunction at 1 month: n = 36 at 3 months; n = 41 at 6 months. For the presence of sexual dysfunction at 3 months: n = 29 at 6 months. *: Data refer to the last month. Contraceptive methods were classified into four categories: none, barrier (condoms), natural (withdrawal or cycle tracking), and medical (pill, IUD, and implant).
(PDF)

## Author contributions

**Conceptualization:** Léa rouchou, Dounia Baïta, Sophie Regueme.

**Data curation:** Léa rouchou.

**Formal analysis:** Léa rouchou.

**Investigation:** Léa rouchou, Dounia Baïta.

**Methodology:** Léa rouchou, Dounia Baïta, Sophie Regueme.

**Project administration:** Sophie Regueme.

**Supervision:** Dounia Baïta.

**Validation:** Dounia Baïta.

**Writing – original draft:** Léa rouchou.

**Writing – review & editing:** Dounia Baïta, Sophie Regueme, Claude Hocke.

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
