## [Decision Letter · Decision Letter 0]

29 Jan 2026

Dear Dr. rouchou,

Thank you for submitting your manuscript to PLOS ONE. After careful consideration, we feel that it has merit but does not fully meet PLOS ONE’s publication criteria as it currently stands. Therefore, we invite you to submit a revised version of the manuscript that addresses the points raised during the review process.

We look forward to receiving your revised manuscript.

Kind regards,

Mahsa Ghajarzadeh

Academic Editor

PLOS One

**Journal Requirements:**

3. For studies involving third-party data, we encourage authors to share any data specific to their analyses that they can legally distribute. PLOS recognizes, however, that authors may be using third-party data they do not have the rights to share. When third-party data cannot be publicly shared, authors must provide all information necessary for interested researchers to apply to gain access to the data. (https://journals.plos.org/plosone/s/data-availability#loc-acceptable-data-access-restrictions)

4. Please ensure that you refer to Figure 2 in your text as, if accepted, production will need this reference to link the reader to the figure.

5. Please upload a new copy of Figures 1 and 2 as the detail is not clear. Please follow the link for more information:  https://journals.plos.org/plosone/s/figures

6. We note you have included a table to which you do not refer in the text of your manuscript. Please ensure that you refer to Table 2 in your text; if accepted, production will need this reference to link the reader to the Table.

7. Please include captions for your Supporting Information files at the end of your manuscript, and update any in-text citations to match accordingly. Please see our Supporting Information guidelines for more information: http://journals.plos.org/plosone/s/supporting-information....

**Additional Editor Comments:**

Dear authors,

Please find the reviewer's comments below.

Please revise point by point and re-submit the revised version.

Reviewers' comments:

Reviewer's Responses to Questions

**Comments to the Author**

1. Is the manuscript technically sound, and do the data support the conclusions?

Reviewer #1: Yes

2. Has the statistical analysis been performed appropriately and rigorously?

Reviewer #1: Yes

3. Have the authors made all data underlying the findings in their manuscript fully available?

Reviewer #1: No

4. Is the manuscript presented in an intelligible fashion and written in standard English?

Reviewer #1: Yes

Reviewer #1: Title: Impact of Voluntary Termination of Pregnancy on Female Sexual Function: A French Monocentric Longitudinal Study

Article type: Research Article

Summary

This prospective, monocentric longitudinal study evaluates the prevalence and trajectory of female sexual dysfunction over six months following voluntary termination of pregnancy (VTOP), using the validated Female Sexual Function Index (FSFI) diagnostic threshold. The authors report a reduction in sexual dysfunction prevalence and improvement in FSFI scores over time among women completing follow-up, with relationship status and psychological symptoms associated with persistent dysfunction.

The topic is clinically relevant and underexplored, and the use of a validated FSFI cutoff strengthens interpretability. However, very high attrition, limited power at later time points, and potential selection bias substantially constrain causal inference and generalizability. The manuscript is generally well written and transparent about limitations, but several methodological and interpretive issues should be addressed before the work can be considered robust enough for publication.

Major Comments

1. Attrition Bias and Internal Validity (Major Concern)

Loss to follow-up is extreme: only 47 of 186 participants (25.3%) completed the 6-month assessment, and only 24 (12.9%) completed all time points. This raises serious concerns about selection bias and threatens the validity of longitudinal conclusions.

• Although the authors correctly state that missingness is likely MNAR and avoid imputation, the primary conclusion (“sexual dysfunction appeared to improve”) relies entirely on completers, who differ systematically from non-completers (e.g., higher prevalence of sexual violence and anxiety).

• Mixed-effects models do not resolve bias when attrition is informative.

Recommendations:

• Explicitly reframe conclusions as conditional on follow-up completion, including in the abstract.

• Provide a baseline comparison table between completers and non-completers at 6 months, not only descriptive statements.

• Consider sensitivity analyses such as:

o Pattern-mixture or worst-case scenario bounds.

o A conservative assumption that non-completers did not improve.

2. Absence of a True Pre-Pregnancy Baseline

Baseline FSFI measurement occurred after pregnancy discovery but prior to VTOP, not before the unintended pregnancy itself. This complicates causal interpretation.

• Improvements may reflect resolution of pregnancy-related stress, not effects attributable to VTOP or post-VTOP recovery.

• The discussion acknowledges this limitation but still implies recovery “after VTOP.”

Recommendations:

• More clearly distinguish between:

o Effects of unintended pregnancy

o Effects of VTOP

o Natural temporal regression

• Revise causal language throughout (e.g., “following VTOP” → “over time after pregnancy resolution”).

3. Sample Size Justification and Statistical Power

Sample size was determined empirically rather than through formal power calculations.

• While acceptable for exploratory research, this limits confidence in null findings for secondary outcomes.

• Several adjusted analyses may be underpowered, especially at later time points.

Recommendations:

• Explicitly state which analyses are exploratory.

• Avoid overinterpretation of non-significant associations (e.g., violence history, VTOP method).

4. Measurement of Psychological Covariates

Psychological symptoms (sadness, anxiety, guilt, fatigue) are assessed using non-validated, study-specific items rather than standardized scales (e.g., PHQ-9, GAD-7).

• This limits reproducibility and interpretability.

• The term “depressive symptoms” may overstate what was actually measured.

Recommendations:

• Clarify terminology (e.g., “self-reported psychological symptoms”).

• Acknowledge the lack of validated mental-health instruments as a limitation.

• Avoid clinical labeling where diagnostic thresholds were not applied.

5. Generalizability

The study is monocentric, with a high proportion of surgical VTOP (66%), exclusion of women with psychiatric comorbidities, and reliance on internet-based follow-up.

• Results may not generalize to:

o Medical abortion settings

o Women with pre-existing mental health conditions

o Populations with limited digital access

Recommendation: Expand the generalizability discussion and avoid extrapolation beyond similar care settings.

6. Qsuestionnaire validation:

The Methods section should more clearly justify the use of the FSFI diagnostic cutpoint (≤26.55), including its psychometric validation, applicability to the French population, and rationale for its use as a binary outcome defining sexual dysfunction.

7. Discussion Enrichment:

In addition to other points that discussed above and should be considered in the discussion, the Discussion would benefit from enrichment addressing how reliance on this diagnostic cutpoint and on a self-reported questionnaire influences interpretation of prevalence estimates, longitudinal change, and clinical relevance, particularly in the absence of a true pre-pregnancy baseline. Specifically, the authors may wish to discuss (optional):

(i) the distinction between changes in FSFI-defined dysfunction prevalence versus changes in continuous FSFI scores;

(ii) the extent to which observed improvements may reflect resolution of pregnancy-related psychological stress rather than post-VTOP recovery per se; and

(iii) how self-reported sexual function measures may be influenced by concurrent emotional state, relationship context, and recall bias over time.

Minor Comments

1. FSFI Interpretation: Clarify whether domain-level changes exceeded minimal clinically important differences, not just statistical significance.

2. Figures: Figure 2 could benefit from displaying raw numerators (n/N) alongside percentages.

3. Language: Some statements in the Discussion imply causality; revise to observational phrasing.

Strengths

• Prospective longitudinal design.

• Use of a validated FSFI diagnostic threshold, addressing a key gap in prior literature.

• Transparent reporting of limitations and ethical safeguards.

• Clinically relevant focus on sexual health, an often-neglected outcome in VTOP care.

Overall Recommendation

Major Revision

The study addresses an important and underreported aspect of reproductive health and meets PLOS ONE’s scope. However, substantial revisions are needed to tighten causal language, address attrition bias more rigorously, and temper conclusions. With these changes, the manuscript could make a valuable descriptive contribution to the literature.

.

Reviewer #1: **Yes:** Saharnaz NedjatSaharnaz NedjatSaharnaz NedjatSaharnaz Nedjat

---

## [Author Response · Author response to Decision Letter 1]

16 Mar 2026

Dear Editor,

Thank you for your constructive feedback and for the opportunity to revise our manuscript. We have carefully addressed all the suggestions and requirements you mentioned and have made the necessary revisions. Below is a detailed summary of the actions we have taken in response to your comments:

1. Adherence to PLOS ONE Formatting Requirements

We have revised the manuscript to ensure it fully complies with PLOS ONE’s formatting requirements, following the provided submission templates. The document has been reformatted to meet all the required standards, and we have attached the revised version for your review.

2. Incorporation of the Title Page into the Main Document

As requested, we have incorporated the title page into the beginning of the manuscript itself, including the full list of authors and their affiliations. This title page is no longer uploaded as a separate file.

3. Third-Party Data Availability

Regarding the use of third-party data in our study, we have updated the "Data Availability Statement" to clarify that these data can be shared. We are now providing an anonymized version of the dataset in Excel format, which can be requested by other researchers.

4. Reference to Figure 2 in the Text

We have added a reference to Figure 2 in the text, as requested. The reference has been inserted at the appropriate point in the manuscript to guide readers to the figure.

5. Improved Quality of Figures 1 and 2

We have uploaded new versions of Figures 1 and 2 with enhanced resolution, as recommended.

6. Reference to Table 2 in the Text

We have added a reference to Table 2 in the text, as requested. The table is now properly cited in the manuscript to ensure readers can easily access it.

7. Captions for Supporting Information Files

We have included captions for all supporting information files at the end of the manuscript, as per PLOS ONE’s guidelines. Additionally, we have updated the in-text citations to match the new captions.

We hope these revisions meet your expectations and comply with the PLOS ONE submission guidelines. Thank you once again for your valuable feedback. Please do not hesitate to contact us if you require any further information.

Kind regards,

Dr Léa Rouchou

Dear Reviewer

We sincerely thank you for your careful reading of our manuscript and for the constructive feedback, which has helped improve the clarity, rigor, and transparency of our study. Below, we provide a detailed point-by-point response to all comments.

Major Comments

1. Attrition Bias and Internal Validity

Reviewer Comment: Loss to follow-up is extreme, raising concerns about selection bias and internal validity. Recommendations include reframing conclusions, providing baseline comparisons between completers and non-completers, and performing sensitivity analyses.

Author Response:

We have explicitly reframed our conclusions throughout the manuscript, including the abstract, to clarify that results are conditional on follow-up completion.

A baseline comparison between completers and non-completers at 6 months has been added (S2 Table). No significant differences were found, except for a slightly higher prevalence of reported physical violence among respondents (p = 0.020).

A conservative sensitivity analysis was conducted assuming that participants with missing follow-up FSFI data did not change their sexual function status. Logistic mixed-effects modeling, adjusting for age, parity, relationship status, history of violence, and self-reported psychological symptoms, confirmed a significant reduction in sexual dysfunction over time (adjusted OR 0.84; 95% CI 0.75–0.95, p = 0.007).

These analyses are now described in the Methods and Results sections.

2. Absence of a True Pre-Pregnancy Baseline

Reviewer Comment: Baseline FSFI measurement occurred after pregnancy discovery, complicating causal interpretation. Recommendations include clearer language distinguishing effects of pregnancy, VTOP, and natural temporal regression.

Author Response:

We have revised the manuscript to consistently use observational language (e.g., “after pregnancy resolution” instead of implying post-VTOP recovery).

The Discussion now more explicitly distinguishes between the effects of unintended pregnancy, the resolution of pregnancy, and the potential influence of VTOP.

3. Sample Size Justification and Statistical Power

Reviewer Comment: Sample size was empirically determined, limiting confidence in null findings.

Author Response:

We clarified that several analyses are exploratory and have revised the text to avoid overinterpretation of non-significant associations, particularly for secondary outcomes (e.g., history of violence, VTOP method).

4. Measurement of Psychological Covariates

Reviewer Comment: Psychological symptoms were assessed with non-validated items, limiting reproducibility.

Author Response:

Terminology throughout the manuscript has been updated to “self-reported psychological symptoms” to reflect the non-validated nature of these measures.

We have acknowledged this limitation explicitly in the Discussion and avoided clinical labeling where diagnostic thresholds were not applied.

5. Generalizability

Reviewer Comment: Monocentric design and specific participant characteristics limit generalizability.

Author Response:

We expanded the Discussion to explicitly note limitations in generalizing results to medical abortion settings, women with psychiatric comorbidities, and populations with limited digital access.

6. FSFI Questionnaire Validation

Reviewer Comment: Methods should justify use of FSFI cutpoint (≤26.55) and its applicability to the French population.

Author Response:

The Methods now describe the FSFI as a validated 19-item tool in French, assessing six sexual function domains.

The rationale for using the ≤26.55 cutpoint as a binary outcome (sexual dysfunction vs. no dysfunction) is clarified, including its validation and alignment with clinical practice guidelines.

7. Discussion Enrichment

Reviewer Comment: Discuss how reliance on FSFI cutpoint and self-report affects interpretation of prevalence, longitudinal change, and clinical relevance.

Author Response:

The Discussion now emphasizes the absence of a true pre-pregnancy baseline and the potential influence of pregnancy-related stress on observed improvements.

We noted that FSFI scores are validated as categorical diagnostic thresholds rather than continuous change scores; therefore, longitudinal analyses focus on prevalence changes.

Observational language has been used to avoid causal overinterpretation.

Minor Comments

1. FSFI Interpretation

Reviewer Comment: Clarify whether domain-level changes exceed minimal clinically important differences.

Author Response: Updated in Results and Discussion to provide these details.

2. Figures

Reviewer Comment: Figure 2 should display raw numerators (n/N) alongside percentages.

Author Response: Figure 2 has been updated accordingly.

3. Language

Reviewer Comment: Revise causal statements in the Discussion.

Author Response: All statements implying causality have been revised to reflect observational study design.

We sincerely thank you again for your thoughtful comments. All suggestions have been addressed to improve transparency, methodological clarity, and interpretability.

Kind regards,

Dr Léa Rouchou

---

## [Editor Report · Decision Letter 1]

26 Mar 2026

Impact of Voluntary Termination of Pregnancy on Female Sexual Function: A French Monocentric Longitudinal Study

PONE-D-26-01018R1

Dear Dr.Rouchou,

We’re pleased to inform you that your manuscript has been judged scientifically suitable for publication and will be formally accepted for publication once it meets all outstanding technical requirements.

Kind regards,

Mahsa Ghajarzadeh

Academic Editor

PLOS One
---

## [Editor Report · Acceptance letter]

PONE-D-26-01018R1

PLOS One

Dear Dr. rouchou,

I'm pleased to inform you that your manuscript has been deemed suitable for publication in PLOS One. Congratulations! Your manuscript is now being handed over to our production team.

Kind regards,

on behalf of

Dr. Mahsa Ghajarzadeh

Academic Editor

PLOS One